# Radiomics of MRI for the Prediction of the Pathological Response to Neoadjuvant Chemotherapy in Breast Cancer Patients: A Single Referral Centre Analysis

**DOI:** 10.3390/cancers13174271

**Published:** 2021-08-25

**Authors:** Filippo Pesapane, Anna Rotili, Francesca Botta, Sara Raimondi, Linda Bianchini, Federica Corso, Federica Ferrari, Silvia Penco, Luca Nicosia, Anna Bozzini, Maria Pizzamiglio, Daniela Origgi, Marta Cremonesi, Enrico Cassano

**Affiliations:** 1Breast Imaging Division, Radiology Department, IEO European Institute of Oncology IRCCS, 20141 Milan, Italy; anna.rotili@ieo.it (A.R.); federica.ferrari@ieo.it (F.F.); silvia.penco@ieo.it (S.P.); luca.nicosia@ieo.it (L.N.); anna.bozzini@ieo.it (A.B.); maria.pizzamiglio@ieo.it (M.P.); enrico.cassano@ieo.it (E.C.); 2Medical Physics Unit, IEO European Institute of Oncology IRCCS, 20141 Milan, Italy; francesca.botta@ieo.it (F.B.); linda.bianchini@ieo.it (L.B.); daniela.origgi@ieo.it (D.O.); 3Molecular and Pharmaco-Epidemiology Unit, Department of Experimental Oncology, IEO European Institute of Oncology IRCCS, 20139 Milan, Italy; sara.raimondi@ieo.it (S.R.); federica.corso@ieo.it (F.C.); 4Department of Mathematics, DMAT, Politecnico di Milano, 20133 Milan, Italy; 5Center for Analysis Decisions and Society, CADS, Human Technopole, 20157 Milan, Italy; 6Radiation Research Unit, IEO European Institute of Oncology IRCCS, 20141 Milan, Italy; marta.cremonesi@ieo.it

**Keywords:** radiomics, breast cancer, magnetic resonance imaging, neoadjuvant chemotherapy, oncology

## Abstract

**Simple Summary:**

Nowadays, the only widely recognized method for evaluating the efficacy of neoadjuvant chemotherapy is the assessment of the pathological response through surgery. However, delivering chemotherapy to not-responders could expose them to unnecessary drug toxicity with delayed access to other potentially effective therapies. Radiomics could be useful in the early detection of resistance to chemotherapy, which is crucial for switching treatment strategy. We determined whether tumor radiomic features extracted from a highly homogeneous database of breast MRI can improve the prediction of response to chemotherapy in patients with breast cancer, in addiction to biological characteristics, potentially avoiding unnecessary treatment.

**Abstract:**

Objectives: We aimed to determine whether radiomic features extracted from a highly homogeneous database of breast MRI could non-invasively predict pathological complete responses (pCR) to neoadjuvant chemotherapy (NACT) in patients with breast cancer. Methods: One hundred patients with breast cancer receiving NACT in a single center (01/2017–06/2019) and undergoing breast MRI were retrospectively evaluated. For each patient, radiomic features were extracted within the biopsy-proven tumor on T1-weighted (T1-w) contrast-enhanced MRI performed before NACT. The pCR to NACT was determined based on the final surgical specimen. The association of clinical/biological and radiomic features with response to NACT was evaluated by univariate and multivariable analysis by using random forest and logistic regression. The performances of all models were assessed using the areas under the receiver operating characteristic curves (AUC) with 95% confidence intervals (CI). Results: Eighty-three patients (mean (SD) age, 47.26 (8.6) years) were included. Patients with HER2+, basal-like molecular subtypes and Ki67 ≥ 20% presented a pCR to NACT more frequently; the clinical/biological model’s AUC (95% CI) was 0.81 (0.71–0.90). Using 136 representative radiomics features selected through cluster analysis from the 1037 extracted features, a radiomic score was calculated to predict the response to NACT, with AUC (95% CI): 0.64 (0.51–0.75). After combining the clinical/biological and radiomics models, the AUC (95% CI) was 0.83 (0.73–0.92). Conclusions: MRI-based radiomic features slightly improved the pre-treatment prediction of pCR to NACT, in addiction to biological characteristics. If confirmed on larger cohorts, it could be helpful to identify such patients, to avoid unnecessary treatment.

## 1. Introduction

The markers currently used in patients with breast cancer to differentiate subtypes or predict treatment responses are traditionally derived from the analysis of a tissue sample, via biopsy or surgery. Radiomics, through the conversion of standard digital imaging into mineable, quantitative data expressing different tumor properties, has gained recognition as a new tool in the field of oncology for noninvasively profiling tumors [1,2]. Particularly, the biological hypothesis driving radiomics research is the potential to enable spatiotemporal and quantitative measurements of both intra- and intertumoral heterogeneity based on medical images, providing the basis for the realization of precision oncology [1,2,3]. The quantification of tumor heterogeneity is crucial indeed, as heterogeneity is a valuable parameter for differentiating between benign and malignant lesions [4], comparing molecular subtypes of breast cancers [5], and determining a patient’s response to neoadjuvant chemotherapy (NACT) [6,7].

The analysis of tumor heterogeneity based on medical imaging can potentially be performed using routinely collected images, such as MRI, without the need for further data collection [2,3]. Moreover, recent studies have pointed out that the responses to NACT in breast cancer patients are associated with radiomic features detected in pre-treatment breast magnetic resonance imaging (MRI) [5,6,7,8,9,10,11,12,13].

Initially, NACT was used to increase the possibility for surgery in inoperable locally advanced or inflammatory breast cancer, but in the last few years it has been increasingly used to treat operable tumors also [14]. Although the ideal outcome is a complete pathological response (pCR) to NACT because of the favorable prognostic value [15], response rates to NACT vary depending on subtype [16], and up to 30% of patients do not benefit from NACT and yet suffer from the toxicity and adverse effects associated with treatment itself [17].

Therefore, there is a need for non-invasive pre-treatment predictors of pCR that can foresee which breast cancer patients will achieve pCR, which will have residual invasive disease (RD), and which will not respond at all. Such predictors would allow for improved stratification of patients into more appropriate treatment regimens, and would prevent delays in effective treatment for patients who would respond poorly.

Recent studies have indeed demonstrated the feasibility and potential benefits of using radiomics in pCR prediction, suggesting that radiomic features extracted from non-invasive imaging examinations, such as pre-treatment MRI, could be associated with the responses (or lack of responses) to NACT in breast cancer patients [7,8,11].

However, the research is still at an early stage and there is both technological and methodological variability in the extraction of the radiological features [2,18]. Therefore, the primary objective of this retrospective study was to determine whether tumors’ radiomic features extracted from a highly homogeneous database of breast MRI could non-invasively predict the response to NACT in patients with breast cancer. The radiomic features were either considered alone or combined with clinical and biological characteristics.

Secondary objectives were to investigate the associations among radiomic features and (a) the four molecular subtypes of breast cancer which were found in our population; (b) the expression of Ki67 by tumor cells.

## 2. Materials and Methods

### 2.1. Study Design

This retrospective study was approved by an institutional review board (approval code: IRB 1926/int/2019), and it was conducted with a consecutive series of patients at a single academic center (a referral center for breast cancer care).

An original approach integrating radiomic features extracted from pre-treatment breast MRI and clinical/biological information was proposed to predict pCR to NACT in our population of biopsy-proven female breast cancer patients. Pathology of surgical specimens was used as the reference standard for the assessment of tumor response to NACT.

### 2.2. Patient Population

Patients with biopsy-proven breast cancer who underwent NACT and breast MRI between 1 January 2017 and 30 June 2019 were extracted from the database of our center according to the following inclusion criteria: (a) breast MRI performed only in our center; (b) NACT planned only in our center; (c) biopsy-proven diagnosis of breast cancer with histological and immunohistochemical analysis performed before NACT; (d) surgery and histopathological analysis of surgery specimens performed after NACT in our center.

Exclusion criteria included (a) negation of patient’s consent to use their data for clinical studies; (b) incomplete or imaging artifacts at MR examination.

### 2.3. MRI Data Acquisition, Imaging Analysis and Segmentation

Image acquisition details of our 1.5 T MRI scanner protocol are reported in the Appendix A [19,20]. The dynamic study consisted of three-dimensional T1-w gradient-echo sequences acquired once before and five times after intravenous administration of 0.1 mmol/kg of a gadolinium chelate at 90 s temporal resolution. The T1-w images were obtained with the following parameters: repetition time 7.39 ms, echo time 3.44 ms, slice thickness 1.4 mm, slice spacing 0.7 mm, field of view 350 × 350 mm^2^. The DICOM images of the first phase after intravenous administration of the gadolinium-based contrast agent were then exported for processing and anonymized.

The software ITK-SNAP version 3.8.0 (http://itksnap.org, accessed on 20 August 2021) was used to assess the tumor’s volume of interest (VOI) [21]. The procedure included, firstly, a manual delineation of the tumor by a dedicated breast radiologist (6 years-experience in breast imaging, and 2 years-experience in radiomics segmentation), and secondly, semi-automatic segmentation based on iterative adaptive thresholding. If deemed necessary, the radiologist made final adjustments manually. The final identified VOI, covering the entire tumor (excluding vessels and hemorrhagic or necrotic foci), was exported in NiFTI format, and it was used to extract the radiomic features for our analysis.

### 2.4. Extraction of Radiomic Features

The package PyRadiomics [22] version 2.2.0 was used to normalize the images and to extract the radiomic features from the VOI identified for each patient. The considered features included morphological, histogram-based, and textural descriptors. Textural features and extraction details are shown in Appendix A.

### 2.5. Pathological Examination

The breast cancer tissue was collected by core needle biopsy before NACT. After NACT, the surgically removed breast tissue was fixed and then embedded in paraffin. The specimens were cut into thin sections and stained with hematoxylin and eosin.

The determination of Ki-67 expression and the assessment of the hormone receptor studies such as estrogen receptor (ER), progesterone receptor (PR), and human epidermal growth factor receptor-2 (HER2) were performed immunohistochemically.

We defined Ki-67 positive staining as Ki-67 staining of 20% or more of cancer cell nuclei and Ki-67 negative staining was defined as Ki-67 staining of fewer than 20% of cancer cell nuclei [23]. Regarding hormone receptor status, we defined tumors with <1% of tumor cells with nuclear staining as ER/PR negative and ≥1% of tumor cells with nuclear staining as ER/PR positive [24].

### 2.6. Determination of Neoadjuvant Treatment Response

The response to NACT was determined based on the final surgical specimen by a breast pathologist with 15 years of experience.

The histopathologic therapeutic response to NACT were classified in two categories based on two different dichotomous criteria: (1) pCR when there was no evidence of RD in the breast or axillary lymph nodes; (2) no pCR or a partial pathological response (namely, residual microscopic foci of cancer cells larger than the median observed in our sample).

### 2.7. Statistical Analysis

Baseline characteristics of patients and tumors are expressed as frequencies and percentages for categorical variables (namely, NACT type; molecular subtype, ER, PR, and HER2; and Ki67 levels with 20% as the threshold value) and as means and interquartile ranges for continuous variables (age, Ki-67 levels).

The associations among patient age, tumor biological characteristics, and pCR to NACT were evaluated by univariate and multivariable logistic regression analysis, and odds ratios (OR) with 95% confidence intervals (CI) were calculated. The multivariable model included age, molecular subtype, and Ki-67 levels (as binary variables), and it was defined as a clinical/biological model. The final multivariable model included only variables (among the three listed above) that were significantly associated with response to NACT.

Statistical analysis of the radiomics workflow is summarized in Appendix A.

A clinical/biological–radiomic model was constructed by multivariable logistic regression analysis, including both variables selected for the clinical/biological model and the radiomic score.

The performances of clinical/biological, radiomic, and clinical/biological–radiomic models were evaluated in terms of area under the receiver operating characteristic (ROC) curve (AUC) with the 95% CI calculated with 2000 bootstrap resampling. Model calibration was formally assessed via Hosmer–Lemeshow test using five intervals to draw the calibration plots.

As a sensitivity analysis, we repeated the previous analysis, considering the percentage of residual microscopic foci of cancer cells as binary variables, but this time using the median as the threshold value (as defined in point 2 of the previous section).

To evaluate the predictions of molecular subtype and Ki-67 levels using radiomic features, we dichotomized the molecular subtype as basal-like vs. all other subtypes (luminal A, luminal B, and HER2+) and Ki67 levels were based on the 20% threshold value.

We replicated the radiomic analyses described for the primary endpoint, both for feature reduction and model construction steps. As for response to NACT, the radiomic scores were calculated as the average (across the repeated folds) predicted probabilities of the models for each of the considered secondary endpoints. Model performance was evaluated by AUC and 95% CI, as previously described.

All the analyses were performed using R 4.0 software [25] and *p* values < 0.05 were considered statistically significant.

## 3. Results

Out of 130 patients treated with NACT who underwent breast MRI during the study period, 83 patients (mean (SD) age: 47.26 (8.6) years) met the inclusion criteria (Figure 1).

Baseline characteristics of the study population and results of the association analysis with responses to NACT are reported in Table 1.

Patients with HER2+ and basal-like molecular subtypes presented a pCR to NACT more frequently compared to patients with luminal A subtype (OR; 95% CI = 28.17; 5.47–144.96 and 9.46; 2.22–40.24,). The association of molecular subtype with response to therapy was also confirmed by higher frequencies of responders among patients with ER and who were PR negative (*p* < 0.0001) and patients without HT (*p* = 0.03). Furthermore, patients with high Ki67 (≥20%) responded more frequently to therapy than patients with low (<20%) Ki67 (OR; 95% CI = 8.92; 1.10–72.56) according to univariate analysis, but the association was not confirmed through multivariable analysis, after adjusting for molecular subtype (Appendix A). The AUC (95% CI) of the clinical/biological model including the four classes of molecular subtypes was 0.81 (0.71–0.90).

Figure 2 shows the segmentation process to obtain the VOIs of the tumors for our radiomics analysis.

A list of the features extracted from the original (not filtered) images is available in Appendix A. The 1037 extracted radiomic features were first reduced to 405 after discharging the ones with near zero variance and with near one correlations with other features. Then they were further reduced to 136 representative features after cluster analysis (Appendix A) performed with a customized in-house function (Appendix A). The top ten radiomic features that contributed most to response to NACT are reported in Table 2 and include one first-order feature (range), considering both the original values and the values obtained after the application of square filter, and seven texture features, one of which (Dependence Entropy) was included both with original values and after applying square filter.

The AUC (95% CI) of the radiomic model was 0.64 (0.51–0.75).

Figure 3 shows the comparison among the ROC curves obtained for the clinical/biological model, the radiomic model, and the clinical/biological–radiomic model.

The AUC (95% CI) of the clinical/biological–radiomic model was 0.83 (0.73–0.92), which was not significantly higher than the AUC of the clinical/biological model alone (*p* = 0.30); otherwise, a significant improvement in the model prediction was observed in comparison to the radiomic model (*p* = 0.0002). All the obtained models were well calibrated (*p* = 1.00, 0.65 and 0.55 for clinical/biological, radiomic, and clinical/biological–radiomic models, respectively). A calibration plot for the complete model is presented in Appendix A.

Using higher and lower percentages of residual microscopic foci of cancer cells as response variables (≥2% and <2%, respectively, where 2% was the median value observed in our sample) did not improve the prediction by radiomic features. Specifically, no radiomic feature was significantly associated with RD in univariate analysis after FDR correction, and the random forest (RF) resulted in an AUC (95% CI) of 0.62 (0.48–0.75).

For molecular subtype analysis, three radiomic features (log-sigma-6-mm-3D_firstorder_Skewness, lbp-2D_glrlm_ShortRunHighGrayLevelEmphasis, and lbp-2D_glszm_LowGrayLevelZoneEmphasis) were found significantly different by molecular subtype after FDR correction in univariate analysis (Appendix A). This result was confirmed through multivariable analysis with the predictor importance levels calculated for these radiomic features, which were among the variables with the ten highest Gini indices (Table 3).

The prediction of the molecular subtype (basal-like vs all other subtypes) according to radiomic features was moderate, with AUC (95% CI) = 0.73 (0.58–0.85) (Figure 4a).

In addition, no radiomic feature was significantly different by Ki-67 (threshold value ≥ 20%) after FDR correction in univariate analysis, and the top ten radiomic features obtained by multivariable RF are reported in Table 3. The prediction of high Ki67 values according to radiomic features was moderate, with AUC (95% CI) = 0.72 (0.60–0.83) (Figure 4b).

Finally, we tested the quality of our radiomic study by applying the radiomics quality score [26], and we obtained an intermediate score level of 19/36 (53%) (Appendix A).

## 4. Discussion

Finding an appropriate biomarker for predicting whether breast cancer patients could achieve pCR with NACT before surgery is a key factor in the assessment of therapy. Currently, the only widely recognized method for evaluating the efficacy of NACT is the assessment of pCR through surgery after patients undergo a therapy regimen [27], and recent studies have shown pCR rates of approximately 40–60% after NACT [12]. Accordingly, delivering NACT to non-responders could expose them to unnecessary drug toxicity and delay access to other potentially effective therapies. Therefore, earlier detection of resistance to NACT is crucial, in order to switch treatment strategies. The current version of the clinical practice guidelines by the “American Society of Clinical Oncology” for women with node-negative estrogenic receptor breast cancer include the use of biomarker tests to help predict whether patients will benefit from NACT [28,29]. MRI driven radiomic studies represent a promising and non-invasive approach. Previous studies proposed prediction models of pCR to NACT in breast cancer based on MRI [8,9,27], and in the last few years, they showed promising results in the prediction of the pCR by extracting radiomics features from pre-NACT breast MRI [2,8,9,10,12,13,30,31].

The AUC (95% CI) of our radiomic model was 0.64 (0.51–0.75), and it was consistent with the radiomics results of analogous studies reported by Braman et al. (AUC = 0.76; 0.69–0.84) [10] and Liu et al. (AUC = 0.64; no CI reported) [9]. It is lower than the ones reported in Xiong et al. (AUC = 0.92; 0.84–0.98) [8] and Zhuang et al. (AUC = 0.82; 0.62–1.00) [13]. Although promising, such results are not accurate enough to propose use in clinical practice as the stand-alone means of predicting pCR to NACT in breast cancer patients.

Although some recent approaches have explored the direct radiomic estimation of response from pre-NACT MRI, these approaches often lack well-understood associations with underlying biological factors [6,7,32,33]. Different molecular subtypes of breast cancer are associated with different sensitivities to NACT in terms of pCR and long-term outcomes [16,34]. Accordingly, in our population, HER2+ and basal-like molecular subtypes showed more likely pCR to NACT compared to the luminal A subtype, and tumors with high Ki67 (≥20%) responded better to NACT than patients with low Ki67 (<20%). Although a high level of Ki-67 expression showed an association with pCR to NACT [35], which may explain the high sensitivity of proliferating tumor cells to NACT, there are discordant results on the role of Ki-67 as a prognostic tool, probably due to different NACT protocols, heterogeneous patient subtypes, and different Ki-67 standards and scoring systems. The AUC (95% CI) of our clinical/biological model, including the four classes of molecular subtypes, was 0.81 (0.71–0.90), thereby confirming that the molecular subtype gives a good prediction of the response to therapy.

A model which includes both clinical/biological and radiomic features may accurately distinguish the non-responders from the responders at an early stage. The AUC (95% CI) of our clinical/biological–radiomic model was 0.83 (0.73–0.92), showing a significant improvement in the model prediction compared to the radiomic model (*p* = 0.0002), though not significantly higher than the AUC of the clinical/biological model alone (*p* = 0.30). Our clinical/biological–radiomic model showed consistent results with other similar combined models reported in the literature, such as the ones of Fan et al. (AUC = 0.71; no CI reported) [7] and Cain et al. (AUC = 0.71; 0.58–0.83) [12]. Xiong et al. [8] and Zhuang et al. [13] reported higher performance.

In univariate analysis, no specific radiomic feature was associated with Ki67, but three radiomic features, namely, *log-sigma-6-mm-3D_firstorder_Skewness*, *lbp-2D_glrlm_ShortRunHighGrayLevelEmphasis*, and *lbp-2D_glszm_LowGrayLevelZoneEmphasis*, were found to be associated with molecular subtype. Notably, these features were not among the top 10 radiomic predictors of response to NACT, suggesting that the radiomic signature is possibly an independent predictor of response to NACT, thereby measuring something different from the molecular subtype.

Concerning a biological interpretation of our results, the radiomic features which are significantly associated with molecular subtype quantify different aspects of signal intensity heterogeneity inside the tumor VOI. This suggests that the early uptake and distribution of contrast medium might behave differently, depending on the molecular subtype of the tumor. Particularly, *skewness* quantifies the asymmetry of the distribution of signal intensities around a mean value, possibly showing different proportions of high- or low-uptake areas for the different molecular subtypes. Moreover, *glrlm_ShortRunHighGrayLevelEmphasis* and *glszm_LowGrayLevelZoneEmphasis* are both texture features which quantify the presence of short runs of high intensity voxels and the distribution of zones with lower intensity voxels, respectively. On the other hand, among the radiomic predictors of response to NACT (Table 2), we found some features that quantify image properties which are different from those related to molecular subtype, such as the degree of randomness observed when comparing the voxel intensities within a neighborhood.

In our study, all patients were recruited in a single institute, and therapeutic choices, MRI image acquisition, and pathological examination were standardized, thereby avoiding confounding factors due to multicenter analyses. First, we chose to consider as responders to NACT only tumors with pCR, differently from some similar recent studies [8,13] which even classified the tumors that showed a RD after NACT as responders (or partial responders) to NACT [36]. Our choice was motivated because a significantly improved prognosis for long-term survival was clearly demonstrated only in breast cancer patients who achieved pCR; and pCR after NACT was the requirement to allow for breast-conserving surgery in some patients initially considered candidates for mastectomy only [37]. Moreover, we performed sensitivity analysis with higher and lower percentages of residual microscopic foci of cancer cells as the responses to give a complete picture of the associations. On the other hand, the pathobiological features of post-NACT RD may be determinants of patient outcome too, and they may differ from the pre-NACT features because resistant tumor cell sub-clones could be selected by therapeutic agents [18]. However, prognostic biomarkers based on post-NACT breast cancer features are limited, and pCR is still considered the only currently validated biomarker of survival [1,27].

Methods of statistical analysis included both univariate and multivariable analyses, with pre-selection of features to reduce feature redundancy, which is a main concern in radiomics analysis. We chose a conservative clustering criterion, resulting in a relatively elevated number of clusters with high intra- and inter-cluster correlations, so by selecting only one feature for each cluster (i.e., the most associated with the outcome), we saved potential important features with small correlations with other features in the same cluster. The results were also robust to different statistical classification approaches: we indeed repeated the model construction with different machine learning methods (Appendix A), obtaining similar results.

This study did not include the assessment of radiomic features’ repeatability and reproducibility. Repeatability studies could not be performed in vivo, since multiple patient acquisitions were not performed, nor in silico, since suitable test objects are not currently available, even if in-house customized breast radiomic phantoms are under development in our group following a previous experience in the pelvis [38].

A reproducibility investigation was not necessary, since the image database used for this study was highly homogeneous in relation to all the acquisition and reconstruction variables, so the extracted radiomic features were considered reproducible within the boundaries of our specific database. Nonetheless, in light of the generalizability and external validation of the model, methodological studies will be necessary to test the reproducibility of the radiomic features included in the radiomic score, and dedicated phantom objects will be useful to this purpose, and for the optimization of intensity scale standardization and magnetic field inhomogeneity correction.

Further limitations included the retrospective nature of the study and the limited number of patients, which avoided us having to split our database into a training and a validation set. For small samples, such splitting is not generally recommended, as it is dependent on just one train–test split [39]. Accordingly, we applied internal repeated k-fold cross validation (5 × 10 CV), which is the best method for training a model on multiple train–test splits, rather than a single one [40]. This gave a better indication of how well the model will perform on unseen data. However, it is important to note that our findings should then been validated in further external study, to generalize and validate them.

Moreover, we analyzed characteristics extracted from a segmentation of the tumor lesion only, without considering the peritumoral tissue. The extraction of radiomic features from the peritumoral region may include other predictive outcome characteristics, such as angiogenic and lymphangiogenic activity or infiltration [7,10,34]. We are going to investigate those characteristics by assessing the variations in model performance when built-in radiomic features extracted from different isotropic expansions of the VOIs are used for the analysis.

## 5. Conclusions

In conclusion, MRI-based radiomic features, when associated with clinical and biological data, slightly improved the pre-treatment prediction of pCR to NACT. Further studies are needed to assess the role of radiomics in the selection of breast cancer patients suited for NACT, to avoid unnecessary treatment.

## Figures and Tables

**Figure 1 cancers-13-04271-f001:**
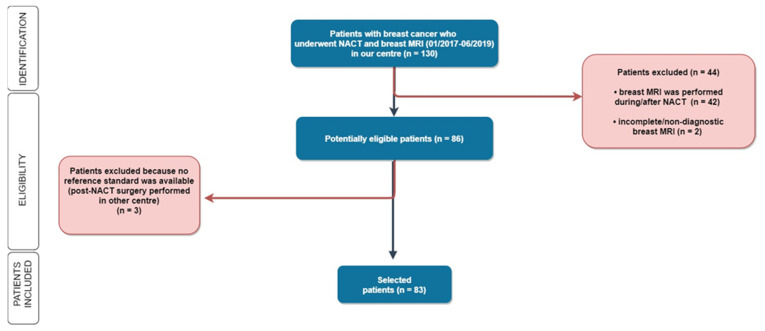
Flowchart. Out of 130 women enrolled in our study, 46 patients were subsequently excluded for the following reasons: for 40 patients the MRI was not performed before the chemotherapy but during or after NACT; for 4 patients the MRI was not technically adequate (1 due to lack of contrast medium administration, 2 due to movement artifacts, 1 due to interrupted examination); and 3 patients underwent post-NACT surgery in another hospital. *MRI = magnetic resonance imaging; NACT = NACT: neoadjuvant chemotherapy*.

**Figure 2 cancers-13-04271-f002:**
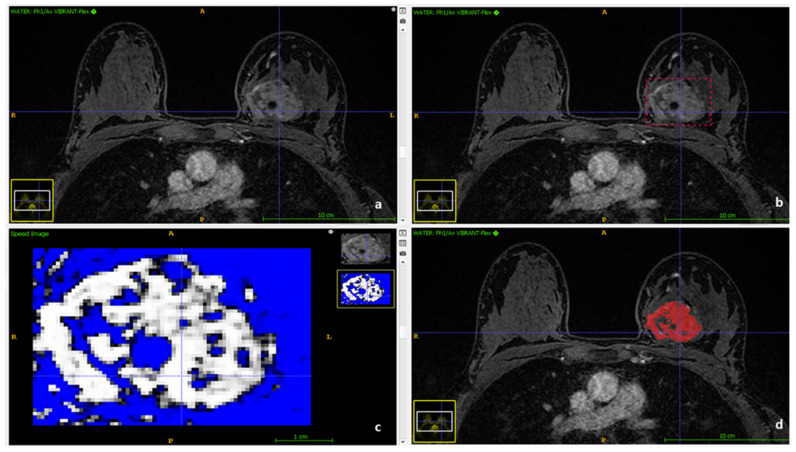
Segmentation process using ITK-SNAP. (**a**) Contrast-enhanced water suppressed T1-weighted series imported in ITK-SNAP. (**b**) Manual delineation of tumor ’s volume by a dedicated breast radiologist. (**c**) Semiautomatic segmentation based on iterative adaptive thresholding. (**d**) The final volume of interest (VOI).

**Figure 3 cancers-13-04271-f003:**
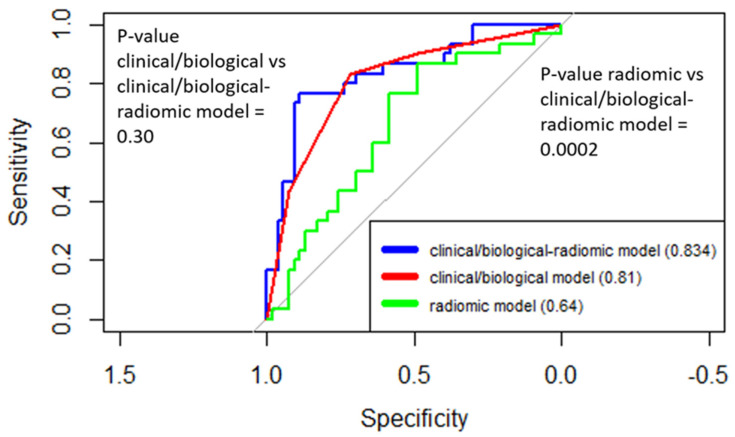
ROC curves for prediction of response to therapy according to clinical/biological, radiomic, and clinical/biological–radiomic models. The clinical/biological model included molecular subtype as an independent variable.

**Figure 4 cancers-13-04271-f004:**
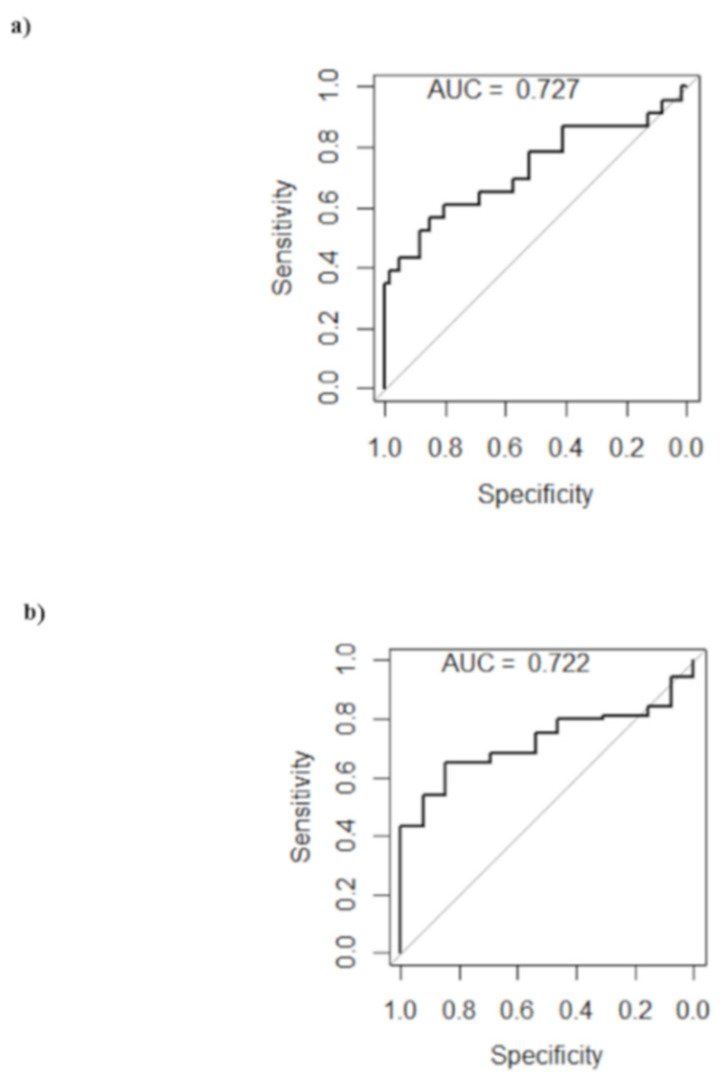
ROC curves. ROC curves for prediction of (**a**) molecular subtype and (**b**) Ki67 according to radiomic models.

**Table 1 cancers-13-04271-t001:** Baseline characteristics of the study population and association analysis with response to therapy.

Characteristics	Overall Cohort (*n* = 83) *n* Patients (%)	Responders (*n* = 30) *n* Patients (%)	Non-Responders (*n* = 53) *n* Patients (%)	*p*-Value *	OR * (95% CI)
Age ^	47.3 (41.1–53.7)	46.5 (41.07–53.8)	47.6 (41.2–52.6)	0.60	0.99 (0.94–1.04)
NACT Type CT CT + HT	65 (81) 15 (19)	28 (97) 1 (3)	37 (73) 14 (27)	Reference 0.03	1.00 (Reference) 0.09 (0.01–0.76)
Molecular Subtypes Luminal A Luminal B Her2-positive Basal-like	29 (35) 14 (17) 17 (20) 23 (28)	3 (10) 2 (7) 13 (43) 12 (40)	26 (49) 12 (23) 4 (8) 11 (21)	Reference 0.71 <0.0001 0.002	1.00 (Reference) 1.44 (0.21–9.81) 28.17 (5.47–144.96) 9.46 (2.22–40.24)
ER Positive Negative	44 (53) 39 (47)	6 (20) 24 (80)	38 (72) 15 (28)	Reference <0.001	1.00 (Reference) 10.13 (3.46–29.72)
PR Positive Negative	41 (49) 42 (51)	3 (10) 27 (90)	38 (72) 15 (28)	Reference <0.001	1.00 (Reference) 22.80 (6.01–86.56)
HER2 Positive Negative	31 (37) 52 (63)	15 (50) 15 (50)	16 (30) 37 (70)	Reference 0.08	1.00 (Reference) 0.43 (0.17–1.09)
Ki67 ^ ≥20% <20%	42.3 (25–60) 13 (16) 68 (82)	50.9 (35–67.5) 29 (97) 1 (3)	37.2 (21.5–45) 39 (73) 12 (23)	0.01 Reference 0.04	1.03 (1.01–1.05) 1.00 (Reference) 8.92 (1.10–72.56)

ER = estrogen receptor; CI = confidence Interval; CT = chemotherapy; HT = hormone therapy, NACT = neoadjuvant therapy; OR = odds ratio; PR = progesterone receptor; * univariate logistic regression model; ^ mean (interquartile range).

**Table 2 cancers-13-04271-t002:** Predicted importance of the top 10 radiomic features obtained from a random forest for prediction of response to therapy.

Feature	Importance (Gini Index)
wavelet-LL_glcm_SumEntropy	1.348
log-sigma-6-mm-3D_glcm_ClusterShade	1.341
squareroot_glcm_ClusterTendency	1.301
original_firstorder_Range	1.269
log-sigma-6-mm-3D_glcm_DifferenceEntropy	1.252
original_gldm_DependenceEntropy	1.207
square_gldm_DependenceEntropy	1.196
original_glrlm_LongRunHighGrayLevelEmphasis	1.154
exponential_glszm_SizeZoneNonUniformity	1.092
square_firstorder_Range	1.073

**Table 3 cancers-13-04271-t003:** Top 10 radiomic variables obtained from the random forest for prediction of molecular subtype and Ki67.

Molecular Subtype		Ki67	
Feature	Importance (Gini Index)	Feature	Importance (Gini Index)
lbp-2D_glrlm_ShortRunHighGrayLevelEmphasis	4.256	squareroot_firstorder_10Percentile	0.608
lbp-2D_glszm_LowGrayLevelZoneEmphasis	2.968	original_firstorder_10Percentile	0.568
original_shape_Flatness	2.550	lbp-2D_firstorder_90Percentile	0.555
lbp-2D_glszm_SizeZoneNonUniformityNormalized	2.453	lbp-2D_glcm_DifferenceEntropy	0.552
lbp-2D_glcm_ClusterProminence	1.967	wavelet-LL_glcm_Imc2	0.534
lbp-2D_glszm_ZoneEntropy	1.847	logarithm_ngtdm_Strength	0.503
wavelet-LH_glrlm_LongRunHighGrayLevelEmphasis	1.693	wavelet-HH_glcm_MCC	0.499
log-sigma-6-mm-3D_firstorder_Skewness	1.494	exponential_glcm_InverseVariance	0.483
wavelet-LH_firstorder_Range	1.362	log-sigma-6-mm-3D_glrlm_ShortRunHighGrayLevelEmphasis	0.478
log-sigma-6-mm-3D_glszm_LargeAreaLowGrayLevelEmphasis	1.347	log-sigma-6-mm-3D_firstorder_Skewness	0.473

## Data Availability

The data presented in this study are available on request from the corresponding author. The data are not publicly available due to privacy concerns, in accordance with GDPR.

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
