# Peer review of "Radiomics of MRI for the Prediction of the Pathological Response to Neoadjuvant Chemotherapy in Breast Cancer Patients: A Single Referral Centre Analysis"

_cancers, 2021, doi:10.3390/cancers13174271_

Round 1

Reviewer 1 Report

The authors have addressed the comments accordingly.

Reviewer 2 Report

I thanks the authors for revising the manuscript according to the suggestions received. I do not have any further comment.

This manuscript is a resubmission of an earlier submission. The following is a list of the peer review reports and author responses from that submission.

Round 1

Reviewer 1 Report

The Authors report on retrospective analysis, aiming at exploring whether radiomic features extracted from breast MRI perfomed on breast cancer patients undergoing primary systemic therapy could be able to predict the likelyhood to undergo pathological complete response after definitive surgery. The article is of interest and generally scientifically sound.

However, as required by the scientific community working on radiomics, the external validation of the model is to be considered mandatory for the generalizability of the findings. Please validate the radiomic model on an external dataset or split your dataset into training and validation cohort to comply with TRIPOD criteria.

Author Response

Dear Reviewer,

Many thanks for your comments. We agree that external validation is an important step to confirm and generalize the findings. As there is not external dataset to use as a validation set for this purpose, we had considered the possibility to split our dataset into training and validation cohort. However, the current literature seems to not recommend such approach, especially for small samples (see Steyerberg EW. Validation in prediction research: the waste by data splitting. J Clin Epidemiol 2018, 103:131-133). Split-sample is dependent on just one train-test split indeed: accordingly, this method is score dependent on how the data is split into train and test sets. For this reason, we had decided to apply an internal k-fold cross validation (5X10). This method is usually the best way to give the model the opportunity to train on multiple train-test splits, rather than a single one, giving   a better indication on how well the model will perform on unseen data (see J. D. Rodriguez et al. Sensitivity Analysis of k-Fold Cross Validation in Prediction Error Estimation. IEEE Transactions on Pattern Analysis and Machine Intelligence 2010, 32,3:569-575 and Ji Eun Park J.E. et al. Reproducibility and Generalizability in Radiomics Modeling: Possible Strategies in Radiologic and Statistical Perspectives Korean J Radiol. 2019 20(7):1124-1137).

In order to better clarify these points, two new paragraphs have been added to the discussion section of the revised version of the paper, as follows:

 “(…) At univariate, no specific radiomic feature was associated with Ki67, while three radiomic features, namely log-sigma-6-mm-3D_firstorder_Skewness, lbp-2D_glrlm_ShortRunHighGrayLevelEmphasis, and lbp-2D_glszm_LowGrayLevelZoneEmphasis, were found to be associated with molecular subtype. Notably, these features were not among the top 10 radiomic predictors of response to NACT, suggesting that the radiomic signature is a possible independent predictor of response to NACT, thus measuring something different from the molecular subtype.”

 and

 “(…) Further limitations included the retrospective nature of the study and the limited number of patients, which avoided us to split our database into a training and a validation set. Especially for small samples, such choice is not generally recommended indeed as it is dependent on just one train-test split [39]. Accordingly, we applied an internal repeated k-fold cross validation (5X10 CV) which is the best method to train the model on multiple train-test splits, rather than a single one [40]. This gives a better indication on how well the model will perform on unseen data. However, it is important to note that our findings should then been validated in further external study, to generalize and validate them.”

 We hope to have appropriately answered to your concerns and that the paper revised following your comments and the other reviewers’ suggestions finds your standard of quality and it is worth of publication in Cancers.

Reviewer 2 Report

The authors have submitted a manuscript covering an important and innovative field, the use of radiomics in early breast cancer. However, the submitted data only add marginally to the body of evidence in the current form. The authors demonstrate that tumor biology and Ki67 are predicitve factors for pCR, which is actually standard of care. They report the retrospective results of a radiomic analysis of MRIs before NACT and after biopsy. A biological hypothesis is completely missing. Furthermore the biological role of the radiomic parameters is not explained at all. In the development of a biomarker a biological hypothesis is at least as important as the pure statistical result. The authors are reporting AUC-results and conclude that the combined radiomic and clinical/biological set of parameters adds to the clinical and biological dataset alone. The AUC changes from 0.81 to 0.834. This is definitely no clinically meaningful additional Information. The AUC for the radiomic parameters alone is 0.64 and definitely insufficient. The question is if a dataset of 83 patients can add anything to the development of a new biomarker at all. A possible way of making use of the data is a formal correlation analysis of clinical/biological and radiomic parameters. This way maybe a hypothesis could be generated if the radiomic data are measuring something different from the clinical/biological data. In summary: For publication I would expect the biological hyopthesis, the detailed description of what is measured in the radiomics dataset and the explanation why this is in line with the biological hypothesis and finally the correlation analysis.

Author Response

Dear Reviewer,

We are grateful for the accurate review of our paper and for your valuable comments.  We tried to solve all the specific comments you rightly pointed out, hoping to have improved the quality of our manuscript according to your points.

 First, we have better discussed in the revised version of the paper a biological interpretation of the features significantly associated with the main investigated clinical/biological parameters (i.e. molecular subtype and Ki-67), and the radiomic predictors of therapy response.  Particularly, in the introduction, we included the following paragraph:

 Particularly, the biological hypothesis driving radiomics research is the potential to enable a spatiotemporal and quantitative measurement of both intra- and intertumoral heterogeneity based on medical images, providing the basis for the realisation of precision oncology [1-3]. The quantification of tumour heterogeneity is crucial indeed as heterogeneity measures is a valuable parameter to differentiate between benign and malignant lesions [4], compare molecular subtypes of breast cancers [5], and determine patient's response to neoadjuvant chemotherapy (NACT) [6,7].

 Second, the results of such association analysis were formally reported in Supplementary Table 7 for molecular subtype and in the text for Ki67, for which no significant association was found, as we reported like the following:

 In addition, no radiomic feature resulted significantly different by Ki-67 (threshold value ≥ 20%) after FDR correction at univariate analysis).

 Third, we have highlighted in the revised version of the paper that the three radiomic features significantly associated with molecular subtype at univariate analysis were not among the top 10 radiomic predictors of response to therapy, suggesting that the radiomic signature is a possible independent predictor of response to NACT. Accordingly, we added the following paragraph:

 At univariate, no specific radiomic feature was associated with Ki67, while three radiomic features, namely log-sigma-6-mm-3D_firstorder_Skewness, lbp-2D_glrlm_ShortRunHighGrayLevelEmphasis, and lbp-2D_glszm_LowGrayLevelZoneEmphasis, were found to be associated with molecular subtype. Notably, these features were not among the top 10 radiomic predictors of response to NACT, suggesting that the radiomic signature is a possible independent predictor of response to NACT, thus measuring something different from the molecular subtype.

Finally, we included also this paragraph at the beginning of discussion:

Currently, the only widely recognized method to evaluate the efficacy of NACT is the assessment of pCR through surgery after patients underwent a therapy regimen [27] and recent studies have shown pCR rates of approximately 40-60% after NACT [12]. Accordingly, delivering NACT to not-responder patients could expose them to unnecessary drug toxicity with delayed access to other potentially effective therapies. Therefore, earlier detection of resistance to NACT is crucial to switch treatment strategy.

We really hope to have appropriately answered to your concerns and that the paper revised following your valuable comments (together with the other reviewers’ suggestions) finds your standard of quality and it is worth of publication in Cancers.